# Involvement of an IgE/Mast cell/B cell amplification loop in abdominal aortic aneurysm progression

Alexia Loste[1,2], Marc Clément[1,2], Sandrine Delbosc[1,2], Kevin Guedj[1,2], Jean Sénémaud[1,2,3], Anh-Thu Gaston[1,2], Marion Morvan[1,2], Guillaume Even[1,2], Grégory Gautier[2,4], Alexander Eggel[5], Michel Arock[6], Emanuele Procopio[1,2], Catherine Deschildre[1,2], Liliane Louedec[1,2], Jean-Baptiste Michel[1,2], Lydia Deschamps[7], Yves Castier[4], Raphaël Coscas[1,8], Jean-Marc Alsac[9], Pierre Launay[2,5], Giuseppina Caligiuri[1,2,10], Antonino Nicoletti[1,2], Marie Le Borgne[1,2]*

1 Université Paris Cité and Université Sorbonne Paris Nord, INSERM, LVTS, Paris, France, 2 DHU FIRE, Paris, France, 3 Department of Vascular and Thoracic Surgery, AP-HP, Bichat Hospital, Université Paris Cité, Paris, France, 4 INSERM UMRS 1149, Centre de Recherche sur l'Inflammation (CRI), Université Paris Cité, Paris, France, 5 Department for BioMedical Research, University of Bern, Bern, Switzerland, 6 Department of Biology and CNRS UMR8113, Ecole Normale Supérieure de Paris-Saclay, Saclay, France, 7 Department of Pathology, AP-HP, Bichat Hospital, Université Paris Cité, Paris, France, 8 Department of Vascular Surgery, AP-HP, Ambroise Paré University Hospital, Université Paris Cité, Boulogne-Billancourt, France, 9 Department of Vascular Surgery, AP-HP, Hôpital Européen Georges Pompidou, Université Paris Cité, Paris, France, 10 Department of Cardiology, AP-HP, Bichat Hospital, Université Paris Cité, Paris, France

* marie.le-borgne-moynier@inserm.fr

**Data Availability Statement:** All relevant data are within the paper and its Supporting information files.

## Abstract

### Aims

IgE type immunoglobulins and their specific effector cells, mast cells (MCs), are associated with abdominal aortic aneurysm (AAA) progression. In parallel, immunoglobulin-producing B cells, organised in tertiary lymphoid organs (TLOs) within the aortic wall, have also been linked to aneurysmal progression. We aimed at investigating the potential role and mechanism linking local MCs, TLO B cells, and IgE production in aneurysmal progression.

### Methods and results

Through histological assays conducted on human surgical samples from AAA patients, we uncovered that activated MCs were enriched at sites of unhealed haematomas, due to subclinical aortic wall fissuring, in close proximity to adventitial IgE+ TLO B cells. Remarkably, *in vitro* the IgEs deriving from these samples enhanced MC production of IL-4, a cytokine which favors IgE class-switching and production by B cells. Finally, the role of MCs in aneurysmal progression was further analysed *in vivo* in ApoE[-/-] mice subjected to angiotensin II infusion aneurysm model, through MC-specific depletion after the establishment of dissecting aneurysms. MC-specific depletion improved intramural haematoma healing and reduced aneurysmal progression.

**Funding:** This work was supported by the Institut National de la Santé et de la Recherche Médicale (INSERM), the Université Paris Cité, and an Emergence grant from the Département Hospitalo-Universitaire 'Fibrosis, Inflammation, REmodeling in cardiovascular, respiratory and renal diseases' (DHU FIRE, Paris, France). AL was supported by the Domaine d'Intérêt Majeur 'Maladies Cardiovasculaires, Obésité, Rein, Diabète' (CORDDIM) from the Region Ile de France, and the Groupe de Réflexion sur la Recherche Cardiovasculaire (GRRC)/Fédération Française de Cardiologie (FFC, Paris, France). The funders had no role in study design, data collection and analysis, decision to publish, or preparation of the manuscript. Powered by.

**Abbreviations:** AAA, abdominal aortic aneurysm; AngII, Angiotensin II; ApoE, apolipoprotein E; DT, diphtheria toxin; MC, mast cell; NAA, non-aneurysmal aortas; RMB, red mast cells and basophil; TLO, tertiary lymphoid organ.

## Conclusions

Our data suggest that MC located close to aortic wall fissures are activated by adventitial TLO B cell-produced IgEs and participate to their own activation by providing support for further IgE synthesis through IL-4 production. By preventing prompt repair of aortic subclinical fissures, such a runaway MC activation loop could precipitate aneurysmal progression, suggesting that MC-targeting treatments may represent an interesting adjunctive therapy for reducing AAA progression.

## Introduction

Arteries are subjected to recurrent mechanical insults which rise from the luminal side of vessels [1]. Vessels have intrinsic capacities to ensure prompt healing of local injuries and are assisted in this task by resident and recruited inflammatory cells [2]. However, the persistence of vascular inflammation can eventually amplify the arterial damage and lead to severe and life-threatening conditions including coronary artery disease, strokes or abdominal aortic aneurysms (AAAs) [3]. Promoting arterial healing by targeting inflammation thus constitutes a major challenge in modern medicine.

In this study, we focused on AAA progression. Monitoring AAA enlargement is problematic due to the discontinuous, so-called 'staccato' growth where month-lasting no-growth/healing periods can be succeeded by a sudden enlargement and again a no-growth/healing period [4]. So far, no pharmacological treatments have been identified to reduce or stop AAA expansion and subsequent arterial rupture, which causes up to 200,000 deaths worldwide each year [5]. The only therapeutic option when the aneurysm diameter exceeds a certain value (55mm in men, 50mm in women) is aorta surgery. Therefore, deciphering the molecular pathways involved in AAA healing and/or progression is essential to set up pharmacological alternatives.

Chronic immune responses involve adaptive and innate immunity and their relationship. In particular, we and others have observed that chronic immune stimulation in AAAs leads to local tertiary lymphoid organ (TLO) development within the adventitia [6–9]. These structures resemble germinal centres from secondary lymphoid organs, as their organization features distinct T- and B-cell areas, lymphatic vessels, high endothelial venules, follicular dendritic cells, and fibroblastic reticular-like cells, both in human samples and mouse model [10]. As in secondary lymphoid organs, TLOs contain germinal centre B cells corresponding to B cells undergoing differentiation into plasma cells [6]. We have reported that TLO development is associated with increased levels of antibodies in the adventitia of human AAAs, in particular IgEs [6], which are also involved in chronic inflammation. Notably, increase in IgE blood concentration correlates with the progression of AAAs and other arterial diseases in patients [11, 12] and *in vivo* studies suggest a pathogenic role for IgEs in experimental AAA [13].

In parallel, triggered by the binding of IgE at their surface, activated mast cells (MCs) release potent proteases, cytokines and vasoactive molecules, such as leukotriene and histamine, directly or indirectly favouring the detersion of the extracellular matrix and a massive loss of the contingent of medial smooth muscle cells, that could eventually lead to aneurysmal expansion [14–16]. Interestingly, MC number increases in diseased arteries [17, 18]. Furthermore, MCs can promote B cell effector functions [19]. As a consequence, MCs have been surmised to be detrimental in AAAs [14, 15, 20]. Studies aiming at inhibiting MC activation with pemirolast, a histamine H1 antagonist, were inconclusive regarding their effect on AAA

reduction [21]. However, pemirolast cannot uncouple IgE-mediated effector processes, which are by far the most potent and specific trigger of MC activation. Hence, to date, the role of MCs in aneurysmal progression remains elusive.

Here, we aimed at investigating the potential mechanism linking MC activity to IgEs and TLO B cell in aneurysmal progression. To do so, we performed histological and *in vitro* assays using human surgical samples from AAA patients. Furthermore, the putative pathogenic role of MCs in aneurysmal growth was evaluated *in vivo*, by inducing MC depletion in Red Mast cell and Basophil (RMB)—ApoE[-/-] mice subjected to angiotensin II infusion during the phase of aneurysm progression.

## Materials and methods

### Mice

Apolipoprotein E-deficient (ApoE[-/-]) mice (RRID:IMSR_JAX:002052) were crossed to RMB (B6.Ms4a2[tm1Mal]) mice [22]. RMB mice carry an additive transgene containing the promoter of the high affinity IgE receptor β subunit controlling the human diphtheria toxin (DT) receptor and the tomato red fluorescent protein [22]. The heterozygous offspring were then intercrossed to generate homozygous ApoE[-/-] RMB mice. Mice were maintained on a C57Bl/6J background and fed a regular chow diet. Induction of AAAs, histology and flow cytometry were performed as described below. All investigations on mice conformed to the Directive 2010/63/EU of the European Parliament, and review and approval of the study was obtained from the Comité d'Ethique Paris Nord #121 (APAFIS #12027).

### Human samples

AAA tissues and blood were obtained from patients undergoing surgery and enrolled in the RESAA (REflet Sanguin de l'évolutivité des Anévrysmes de l'aorte abdominale) and FAD (Fighting Aneurysmal Disease) studies from 2007 to 2020 [23]. As expected [5], AAA patients were predominantly men, and presented risk factors such as age, hypertension, hyperlipidaemia and smoking (S1 Table). All patients gave written informed consent, and the protocol was approved by the Comité Consultatif de Protection des Personnes dans la Recherche Biomédicale (CCPPRB, Paris-Cochin, approval no. 2095 from September 23[rd] 2003) and by the Comité de Qualification Institutionnel de l'INSERM (approval no. 01–024). Our study complies with the Declaration of Helsinki. Control aortas were sampled from dead organ donors from 2009 to 2020 with the authorisation of the French Biomedicine Agency (PFS09-007, S2 Table). Depending on their size, samples were cut in several pieces that were used to prepare conditioned medium, and/or digested for flow cytometry analysis, and/or fixed in 3.7% paraformaldehyde (PFA) for histological studies, and stored in the Inserm human CV biobank (BB-0033-00029), included in the European network BBMRI-ERIC. The authors had no access to information that could identify individual participants. Experiments were conducted from 2014 to 2022.

### Histology and immunofluorescence on human samples

PFA-fixed aortic tissues were paraffin-embedded. Four μm-thick sections were deparaffinised in toluene and rehydrated in ethanol. Sections were subjected to Carstair's stain, orcein stain or Perl's + diaminobenzidine (DAB) stain. Alternatively, sections were incubated with retrieval reagent (R&D Systems), then immunostained using antibodies against IgE (goat polyclonal, Vector Laboratories), MC tryptase (rabbit monoclonal, clone EPR8476, abcam; or mouse monoclonal, clone AA1, abcam), CD20 (mouse monoclonal, clone L26, Dako; or goat

polyclonal, Thermofisher), CD117 (rabbit polyclonal, Dako), or glycophorin A (rabbit monoclonal, clone EPR8200, abcam). Immunostaining with isotypes were conducted to verify the specificity of the primary antibodies. Overnight incubation at 4°C with primary antibodies were followed by incubation with fluorophore-coupled anti-species antibodies (Jackson Immunoresearch) for one hour at room temperature. References for antibodies are shown in S1 File. Nuclei were then stained with Hoechst 53542, and slides were mounted with fluorescent mounting medium (ProLong Gold, ThermoFisher). Slides were kept in the dark at 4°C. Blind analysis was performed for the presence of IgE+ TLOs, MCs and haematomas.

## Preparation of conditioned medium and immunodetection of soluble molecules

For the preparation of conditioned medium, the adventitia of AAA and control aortas were separated from the media and cut into small pieces (5 mm$^3$). The samples were then incubated for 24 hours at 37°C in a standardised volume (6 mL/g of tissue) of RPMI 1640 medium (Gibco) supplemented with antibiotics and antimycotics. The conditioned medium was then centrifuged and the supernatant aliquoted and frozen at -80°C until use.

Concentration of cytokines in conditioned medium from adventitial tissues and plasma was analysed by CBA (BD biosciences). Bead fluorescence was recorded on a LSRII flow cytometer (BD). Tryptase was quantified by ELISA (USCN and Invitrogen). Immunoglobulins were analysed using a bio-plex pro human isotyping assay (Bio-Rad) according to manufacturer instructions. Beads were analysed on a Bioplex-200 analyser (Bio-Rad). IL-4 and Tryptase were analysed on a different set of patients than the experiments looking at the effect of conditioned medium on mast cells *in vitro*, as the latest consumed the conditioned medium for several patients, and the leftovers from the other patients had been thawed and frozen repetitively making them unsuitable to conduct cytokine dosage.

## ROSA mast cell line

The ROSA$^{KIT WT}$ human cell line (RRID:CVCL_5G49) was cultured in the presence of recombinant human SCF (80 ng/mL, R&D Systems), as described previously [24]. Cells were stimulated in the presence of conditioned medium from AAA and control adventitia, or PMA (10ng/mL, Sigma-Aldrich) and ionomycin (1μM, Sigma-Aldrich) (hereafter P+I condition) or IgE (2μg/mL, BeckmanCoulter) for 15 minutes followed by anti-IgE antibody (5μg/mL). After 1 hour of stimulation, the expression of CD63 in MCs was analysed by flow cytometry. Alternatively, after 4 hours of stimulation, cells were collected and dry pellets were frozen for gene expression analysis. In some experiments, 5μM DARPin$^®$ protein bi53_79 [25] was added to the conditioned medium. Experiment was repeated twice with similar results.

## Gene expression analysis

Total RNAs were extracted from ROSA MCs using the PureLink RNA mini kit (Invitrogen), and mRNA reverse transcription was performed using iScript reverse transcriptase (Bio-Rad). Real time quantitative PCR (RT-qPCR) was performed on a CFX 100 thermocycler (Bio-Rad) using the following primers: forward IL-4: GCAGTTCTACAGCCACCAT; reverse IL-4: ACTCTGGTTGGCTTCCTTCA. One nanogram of cDNA from each sample was mixed with forward and reverse primers (300 nM) and SYBR Green master mix (Bio-Rad). The amplification program was as follows: A first step of initial denaturation at 95°C for 3 minutes, then 40 cycles of 3 steps: denaturation at 95°C 15 seconds, annealing at 57°C for 15 seconds, and a final extension at 72°C for 30 seconds. The data were analysed using the $2^{-\Delta\Delta Ct}$ formulas: the Ct values of IL-4 were normalised to the average Ct values of RPS18 (forward RPS18:

GCGGCGGAAAATAGCCTTTG; reverse RPS18: GATCACACGTTCCACCTCATC) and ACTB (forward ACTB: TCCCTGGAGAAGAGCTACG; reverse ACTB: TTTCGTGGATGCCACAGGAC) and non-treated MCs were used as reference.

## Ang-II abdominal aortic aneurysms in mice

28-week-old ApoE-RMB males were used for the experiments, as males are more susceptible to develop AAAs after Angiotensin II (AngII) infusion [26]. AngII (#A9525, 1 mg/kg/day, Sigma-Aldrich, St Louis, Missouri) was continuously infused into the experimental mice via osmotic mini-pumps (Model 2004, Alzet, Charles River Laboratories) that were surgically implanted subcutaneously in the interscapular region under anaesthesia induced by intraperitoneal injection of ketamine (100 mg/kg) and xylazine (20 mg/kg). Surgery was followed by a buprenorphine intraperitoneal injection (0.1 mg/kg) for analgesia. Three mice (8%) died within the first 10 days. Fourteen days after inducing AAAs through AngII infusion [27], mice received two intraperitoneal injection of DT (1μg/ injection, n = 18) at a one-day interval (or PBS in the control group, n = 17) in order to deplete MCs. Randomization was achieved by having mice from each experimental group in each cage. Mice were sacrificed at day 28 by intracardiac exsanguination under overdose of anaesthesia (intraperitoneal injection of 150 mg/kg ketamine and 30 mg/kg xylazine). Before the exsanguination, a cell suspension from the peritoneum was obtained by peritoneal lavage with 5 mL of ice-cold PBS. Blood was withdrawn from the right ventricle of the heart and collected in EDTA tubes for blood cell analysis. The heart and aorta were dissected, photographed and mounted on cryomolds for further histomorphological analysis based on cryosections. The experiment was repeated twice with similar results. Pooled number of mice from both experiments who survived and developed aneurysms in each group are summarized in S3 Table. DT injection had no effect on peritoneal MCs and aortic tissue MCs in control ApoE$^{-/-}$ mice (Fig 4C and S7 Fig). Concentrations of cytokines in the plasma were measured by Luminex (R&D Systems) and levels of tryptase (mcpt6) were quantified by ELISA (Invitrogen).

## Histology on mouse tissues

MC depletion was analysed on cryosections of mouse aortic roots stained with Toluidine Blue. Images were captured on a Zeiss Axio Observer Z7 inverted microscope, and MCs were counted manually. A detailed blind analysis of aneurysms was performed for mouse abdominal aortas displaying macroscopic evidence of the occurrence of an aneurysm. Cross-cryosections (10 μm) of the aortic segments taken at different levels of the aneurysms (every 300 μm) were stained for collagen with picrosirius red. Images were acquired under polarised light. The size of the different layers (media, haematoma, adventitial fibrous cap) and the extent of collagen deposition were quantitatively assessed using Image J.

## Image acquisition

Images were digitally captured using an AxioObserver epi-fluorescent microscope equipped with a Colibri 7 LED generator (Zeiss) and an ApoTome system and running Zen Software (Zeiss). Macroscopic images of the human samples were acquired using the NanoZoomer Digital slide scanner (Hamamatsu Photonics).

## Flow cytometry analysis

Mouse blood samples collected in EDTA were incubated in Ammonium-Chloride-Potassium lysis buffer for 5 minutes at room temperature to lyse red blood cells. Cells from blood samples

or 1 mL of peritoneal lavage were stained for dead cells with Live/Dead fixable far red Dead cell stain kit (Invitrogen) for 30 minutes at 4˚C, then incubated with a purified rat anti-mouse CD16/32 (FcBlock, BD Biosciences) for 15 minutes at 4˚C. Cells were then incubated for 20 minutes at 4˚C with the following antibodies (from BD Biosciences unless stated otherwise): BV605 rat anti-mouse CD19 (clone 1D3), FITC rat anti-mouse CD117 (clone 2B8), PerCP rat anti-mouse CD45 (clone 30-F11), APC hamster anti-mouse FcεRIα (clone MAR-1, Biolegend), AF700 rat anti-mouse CD3 (clone 172A). References for antibodies are shown in S1 File.

For human tissue analysis, fresh adventitial layer samples were weighed, cut into small pieces (<1 mm) and digested using a previously described protocol [28]. After a wash step, the cells were stained for dead cells with Live/Dead fixable far red Dead cell stain kit (Invitrogen) for 30 minutes at 4˚C, then incubated for 20 minutes at 4˚C with a combination of the following mouse anti-human antibodies: AF488 anti-CD3 (clone UCHT1, BD Biosciences), PE anti-CD203c (clone NP4D6, Biolegend), PerCP anti-FcεRIα (clone AER-37, Biolegend), BV421 anti-CD117 (clone 104D2, BD Biosciences), BV500 anti-CD45 (clone HI30, BD Biosciences), PE-Cy7 anti-CD63 (clone H5C6, BD Biosciences), AF700 anti-CD19 (clone HIB19, BD Biosciences), AF647 anti-CD107a (clone H4A3, BD Biosciences), BV421 anti-CD27 (clone O323, Biolegend), BV570 anti-HLA-DR (clone L243, Biolegend), BV785 anti-CD19 (clone HIB19, Biolegend), APC anti-IgD (clone IA6-2, BD Biosciences), FITC anti-CD38 (clone HIT2, BD Biosciences), PE anti-CD45 (clone HI30, BD Biosciences), BV605 anti-CD45 (clone HI30, BD Biosciences).

For mouse peritoneal cells and human adventitial cells, flow-count fluorospheres (Beckman Coulter) were added to the samples before the acquisition on the cytometer in order to calculate absolute count of cells. Data were acquired on a LSRII flow cytometer or an ARIA III cell sorter (BD Biosciences) and analysed using FACSDiva (BD Biosciences) and FlowJo (TreeStar) softwares.

## Statistical analysis

Statistical analysis was performed using the JMP9 and GraphPad Prism 9.0 software. To compare two populations, we used Student t-tests or non-parametric Mann-Whitney tests when sample size was too small (n<10) or when values were not normally distributed. Comparison of more than two populations was performed with Kruskal Wallis test followed by Dunn's multiple comparison. Paired values were compared using Wilcoxon matched-pairs signed rank test. Pearson correlation coefficients or p-values of chi-square contingency tests were calculated to assess correlation between continuous or categorical variables, respectively. A mixed model that uses a compound symmetry covariance matrix was fitted in GraphPad Prism using Restricted Maximum Likelihood (REML) to analyse effects of two categorical variables with repeated measures on quantitative variables. Quantitative data are expressed as mean +/- standard error.

## Results

### IgE producing-TLOs are associated with aneurysmal wall unhealed fissures in patients

Tissue infiltrated IgE were enriched in aneurysmal samples and soluble IgE plasma levels were elevated in AAA patients compared to control non-aneurysmal aortas (NAAs, S1 Fig). In parallel, TLOs were found in a majority of AAA samples (20 out of 25, Fig 1A) whereas they were absent in all NAA, in agreement with previous findings [6–9]. In particular, adventitial TLOs

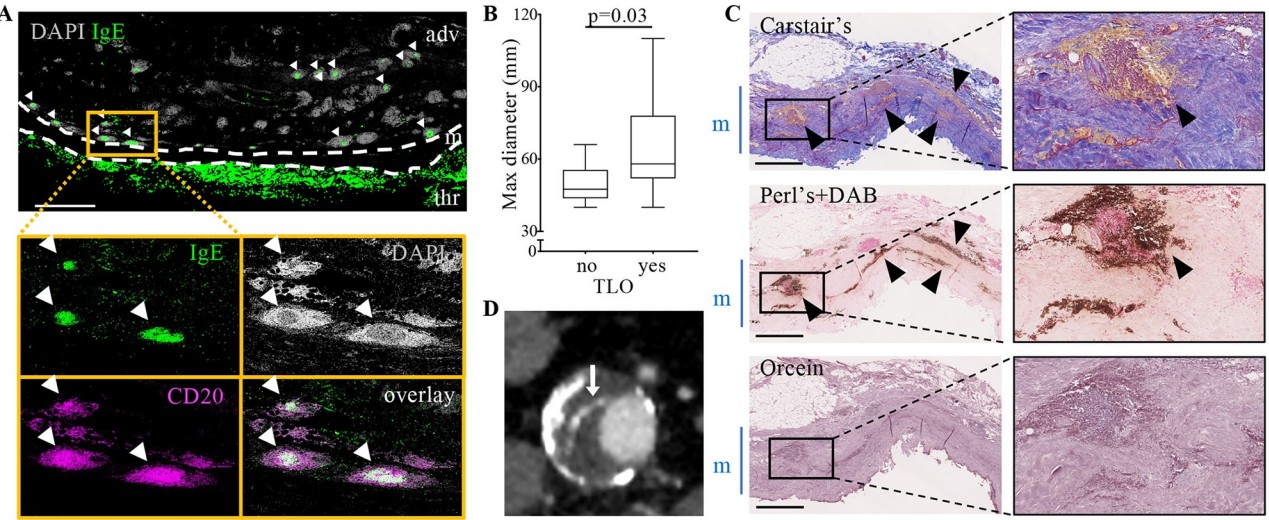

**Fig 1. IgE-producing TLOs are present in the adventitia of human AAAs.** (A) Cross-sections of AAAs were stained for CD20 (magenta) and IgE (green). Dense DAPI$^+$ lymphoid aggregates highly enriched in CD20+ B cells correspond to TLOs. Several TLOs displayed a strong IgE staining projecting in the CD20+ B cells comprised in the DAPI "light zone" of TLOs (white arrowheads). The magnified inset shows 5 adventitial TLOs (cluster of DAPI+ CD20+ B cells), 3 of which are IgE+. adv: adventitia, m: media, thr; thrombus. Representative image of 9 different samples. Scale bar: 2.5 mm. (B) Maximum diameter size of AAAs depending on the absence (no) or the presence of TLOs. p value (Mann-Whitney test) is indicated on the plot. (C) Serial cross-sections of AAAs were coloured with Carstairs' staining, where fibrin appears in bright pink, collagen in blue, red blood cells in yellow and nuclei in dark violet. Perl's+DAB staining highlight redox-active iron in dark brown, nuclei have been counterstained in nuclear fast red. Orcein staining permits to detect elastin fibres in dark violet. The lumen is at the bottom of each picture and the media (m) is indicated by the blue bar on the side of the pictures. Black arrowheads point at red blood cells outside blood vessels, reminiscent of recent intramural haematomas. Scale bar: 1 mm. Right panels: magnified insets. (D) Contrast-enhanced tomography angiograms of an AAA displaying blood disruption from the aortic lumen to the aortic wall through the intraluminal thrombus (white arrow).

were consistently found in samples from patients with larger aneurysms (Fig 1B), suggesting their higher frequency in these patients. In these adventitial TLOs, the proportion of germinal centre B cells was dramatically increased (S2 Fig), suggesting that adventitial TLO B cells in large aneurysms are engaged into an Ab-producing program from which diverse Ig isotypes could emerge. Interestingly, we observed a strong staining for IgEs within the TLO "light zone" (where the centrocytes, i.e. B cells differentiating in immunoglobulins-producing cells, are located [29]; Fig 1A) in 45% (n = 9) of the samples. Altogether these results suggest that germinal centre B cells within adventitial TLOs situated within large AAAs may actively produce IgE immunoglobulins.

Intriguingly, combined histological analysis of Carstair's, Perl's + DAB and Orcein staining revealed the presence of recent or past wall fissuring with haematoma formation in the media in 92% AAA samples (Fig 1C and S3 Fig and Table 1). Indeed, 84% of total AAA tissues (n = 21) presented an accumulation of red blood cells (Fig 1C), reflecting the recent occurrence of intramural haematomas. Of note, the diameter of the aneurysmal samples was significantly larger than 3 cm, implying a greater tension and mechanical stress, than the one of the non-aneurysmal aortas (diameter < 3 cm) [5] and an attentive evaluation of pre-surgery tomography angiograms consistently revealed the presence of at least one detectable aneurysmal wall macroscopic fissure, with radiologic contrast penetrating *via* blood disruptions from the aortic lumen to the aortic wall through the intraluminal thrombus, in 78% of patients with intramural haematomas (Fig 1D and S3D Fig). In some samples, we were able to observe the entry site of blood from the lumen on AAA histological sections, suggesting that the fissures leading to intramural haematomas were provoked by tears (micro-fissures) initiated from the aorta

**Table 1. Patients' characteristics according to the presence of intramural haematomas.**

| | None (n = 2, 8%) | Ancient (n = 2, 8%) | Recent (n = 21, 82%) |
|---|---|---|---|
| Age (years) | 70 +/- 7 | 67+/- 1 | 71 +/- 2 |
| Male (%) | 100% | 100% | 90% |
| Max. aortic diameter (cm) | 63 +/- 3 | 66 +/- 14 | 61 +/- 4 |
| Diabetes (%) | 0% | 0% | 7% |
| Hypertension (%) | 100% | 100% | 87% |
| Hyperlipidaemia (%) | 0% | 50% | 53% |
| Smoking (%) | 50% | 50% | 93% |

Values are mean +/- SEM or %. Ancient intramural haematomas are characterised by ferrous iron deposits with few red blood cells in the media (S3B Fig) and recent intramural haematomas are characterized by abundant red blood cells in the media (Fig 1C).

lumen (S4A, S4B, S4D and S4E Fig). Unhealed intramural haematomas could also be evidenced as large areas of ferrous iron deposits next to modest red blood cell accumulation in the media (S3B Fig) in 8% (n = 2) of samples. Importantly, IgE+ TLOs were frequently localized near these intramural haematomas (Fig 2A–2C), suggesting that the maturation of B cells towards IgE-producing cells within adventitial TLOs and the occurrence of micro-fissures are linked.

## MCs are enriched at sites of aneurysmal wall fissures and IgE-producing TLOs in human AAAs

In parallel, we observed that the adventitia of AAA patients was significantly enriched in MCs compared to the adventitia of NAA organ donors (Fig 2E and 2F and S5 Fig), in agreement with previous observations [18]. Moreover, MCs were often located next to IgE+ TLOs (Fig 2B and 2C), and extracellular tryptase+ granules reflecting recent MC activation could be observed around these MCs by immunostaining on AAA sections (Fig 2G and 2H). Indeed, as compared to NAA samples, adventitial MCs in AAA samples were degranulating, as documented by the expression of the degranulation/MC activation marker CD107a analysed by flow cytometry (Fig 2E and 2F). Strikingly, MCs enriched in AAA tissues were consistently found at sites of haematomas on AAA sections (Fig 2D and S4C and S4F Fig) where their granule enzymes could favour arterial wall rupture. Altogether, these results suggest that MCs and IgE+ TLOs could concur and be linked together with the occurrence of arterial wall injuries and aneurysmal progression.

## Soluble molecules from human AAA adventitia induce MC degranulation and IgE-dependent production of IL-4

Given that MCs displayed an activated phenotype only in the adventitia of AAA patients (Fig 2), we assessed the putative presence of MC activation triggers in the conditioned medium prepared from the adventitia of AAA as compared to control NAA tissues. We observed that conditioned medium from AAA adventitias induced more surface expression of the degranulating marker CD63 and more IL-4 mRNA transcription than conditioned medium from NAA adventitia in cultured ROSA human MCs (Fig 3A and 3B). This is important because the cytokine IL-4 is required to carry out an IgE antibody class switch recombination and hence allow the production of IgE by B cells [30]. Therefore, these results point at an unforeseen function of MCs in AAAs, whereby MCs could orientate the local adaptive immune response generated in adventitial TLOs toward IgE production. In turn, the binding of adventitial IgE on their

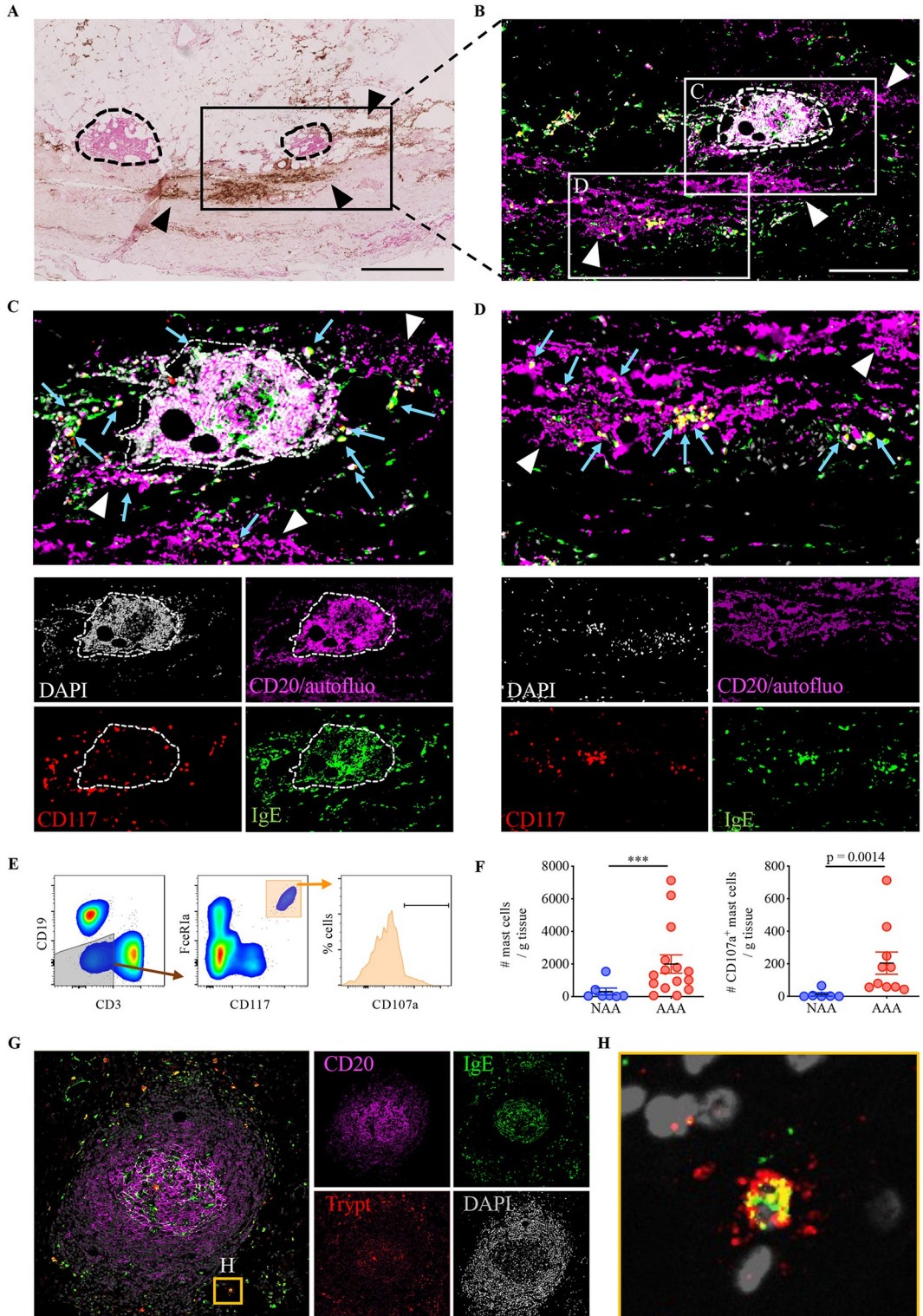

**Fig 2. IgE-producing TLOs, activated MCs and intramural haematomas are in close proximity in human AAAs.** (A) Cross-sections of AAAs were coloured with Perl's+DAB staining and nucleus were counter-stained with nuclear fast red. TLOs (circled with dashed line) are found at proximity of haematomas (arrowheads). Scale bar: 500 μm. (B) Consecutive sections were stained for CD20 (magenta), IgE (green), CD117 (red) and DAPI (grey). B cells (CD20+ and DAPI+) appear in light pink, autofluorescent red blood cells (low intensity in the CD20 channel and DAPI−) appear in magenta, and MCs

(CD117[+] IgE[+]) appear in yellow. Scale bar: 250 μm. (C) Magnified inset highlighting the close proximity of IgE[+] TLOs (dashed circle), red blood cells (white arrowheads) and MCs (blue arrows). (D) Magnified inset highlighting MCs (blue arrows) at proximity to intramural haematoma. (E-F) Adventitia from NAA organ donors and AAA patients were digested and analysed by flow cytometry after the addition of fluorescent count beads. Singlet, not autofluorescent, live CD45[+] cells were selected, and MCs were identified as CD3[-] CD19[-] FcεRIa[+] CD117[+] cells (E). The number of MCs (F, left) and activated (CD107a[hi]) MCs (F, right) was calculated in each sample. Error bars represent mean +/- standard error; Mann-Whitney test (***, p < 0.001). (G-H) Cross-sections of AAAs were stained for CD20 (magenta), IgE (green), and tryptase (red). MCs (tryptase[+] IgE[+]) were detected around TLOs. Scale bar: 100 μm. The magnified inset (H) shows a degranulating tryptase[+] MC at proximity of a TLO.

high affinity receptor on MCs could be responsible for triggering MC activation. To directly assess this hypothesis, we pre-incubated the conditioned medium with anti-IgE DARPin® protein bi53_79 (5 μM) before adding it to MCs (Fig 3C and 3D). DARPin® protein bi53_79 is a soluble molecule that specifically binds to the IgE-Fc portion thereby preventing IgE binding to FcεRI [25]. Whereas treatment of conditioned medium with bi53_79 did not prevent MC degranulation (Fig 3C), it decreased IL-4 mRNA production compared to untreated conditioned medium from AAA tissues (Fig 3D). IgEs contained in the conditioned medium from AAA adventitias therefore emerge as instrumental in promoting MC IL-4 production.

To assess if IL-4 production, MCs, and IgEs are also linked *in vivo*, we measured IgE, tryptase, and IL-4 quantities in the conditioned medium from AAA adventitias. We found a statistically significant positive correlation between the concentration of IgEs and IL-4 (p<0.001), as well as between the concentration of IL-4 and tryptase (p<0.05, Fig 3E and 3F). No such correlations were observed in the plasma of paired patients, whereas concentration of IL-4 correlated with concentration of other inflammatory cytokines such as IL-6 (S6 Fig). Levels of adventitial and circulating IL-4, and adventitial and circulating tryptase, were not correlated (S6 Fig). These observations support the existence of a local amplification loop in the adventitia of AAAs involving IL-4 producing MCs and IgE-producing B cells. Surprisingly, tryptase, IgE and IL-4 levels were also correlated with levels of interferon (IFN)-gamma in the adventitia, which is usually described as an IL-4 antagonist (S6 Fig).

**MCs aggravates aneurysmal progression in mice.** Previous studies have identified MCs as directly involved in provoking arterial damage [14, 15, 20]. Consistently, the presence of medial/adventitial MCs with an activated phenotype was associated to micro-fissures in our human AAA samples, suggesting that MCs could play a role in the aneurysmal remodelling of the aortic wall, upon the occurrence of tissue fissuring. Therefore, we asked whether specific depletion of MCs could prevent the aneurysmal remodelling in ApoE[-/-] mice subjected to chronic infusion of Angiotensin II (AngII), a well-known mouse model of aortic dissection eventually followed by aneurysmal progression [27]. In this setting, aneurysm of the abdominal aorta starts within 10 days upon the formation of large intramural haematomas due to the occurrence of multiple medial tears originating from the lumen, at the origin of the main side branches [31]. In order to conditionally induce specific MC depletion by DT injection after the occurrence of the aortic fissuring, we crossed ApoE[-/-] mice with RMB mice (ApoE-RMB mice) [22]. ApoE-RMB mice were subjected to infusion with AngII for 28 days. We injected DT (or PBS as a control) 14 days after the beginning of AngII infusion. Thus, MCs were depleted after the occurrence of the dissection in this model [27], allowing to evaluate the effect of their depletion on the remodelling of the dissected aortas and their potential effect on aneurysmal progression 14 days later (Fig 4A).

Survival was similar in the two groups (S3 Table). As anticipated, the proportion of mice presenting an intramural haematoma at day 28 was equivalent in DT-treated mice (44%, n = 8/18) and in the control group (47%, n = 7/17). DT injection almost completely depleted

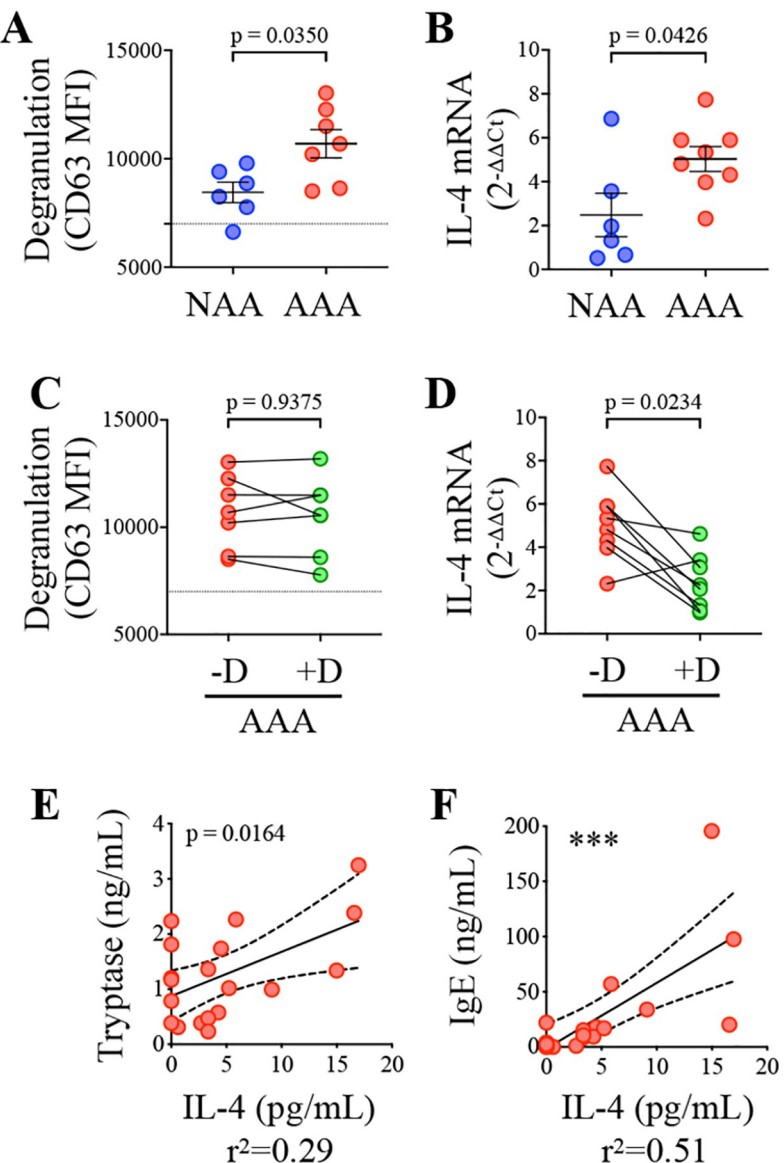

**Fig 3. MC degranulation and IgE-dependent IL-4 production in response to conditioned medium from AAA adventitia.** (A-D) ROSA MCs were cultured in the presence of conditioned medium from adventitia of control organ donors (NAA) and AAA patients. DARPin® protein bi53_79 (5 μM) was added (+D) or not (-D) to inhibit IgE binding to MCs' FcεRI (C-D). (A, C) After 1 hr, degranulation (MFI CD63) was assessed by flow cytometry. The dotted line indicates CD63 MFI in non-stimulated cells. (B, D) After 4 hrs, mRNA IL-4 level was analysed by RT-PCR ($2^{-\Delta\Delta Ct}$, normalised to RP18S and non-stimulated cells). A,B: Mann-Whitney test; C,D: Wilcoxon matched-pairs signed rank test. (E, F) Concentrations of IL-4, tryptase, and IgE were measured in conditioned medium from AAA adventitia. Pearson correlation analysis (***, $p < 0.001$).

MCs from the aortic tissue (Fig 4B and 4C) and the peritoneal cavity (S7 Fig). It should be noted that in RMB mice, DT induces transient depletion of both MCs and basophils. While basophils fully repopulate the bloodstream within 12 days, the repopulation of peripheral tissues such as skin or peritoneum by MCs is much slower [22, 32]. As a result, two weeks after DT administration (Day 28 of AngII infusion), MCs were still completely absent from the

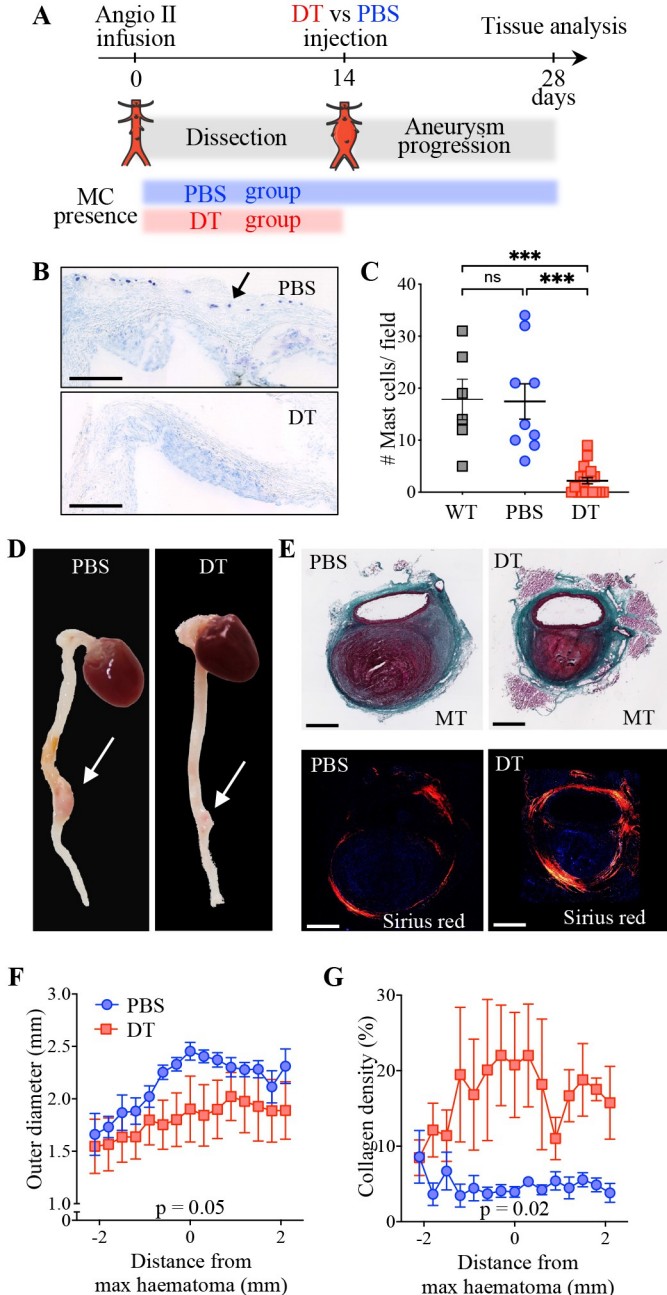

**Fig 4. MC depletion decreases aneurysmal growth after dissections in ApoE-RMB mice.** (A) Experimental protocol, showing the normal course of aneurysm development in the Ang II infusion model, and the presence or absence of MCs depending on the treatment (PBS in blue or DT in red). (B) Toluidine blue stain on aortic root cross-sections of DT- and PBS-treated mice. The arrow points to one of the MCs detected in the aortic root adventitia of a PBS control mouse. Scale bar: 250 μm. (C). Quantification of MCs in aortic roots of DT-treated ApoE$^{-/-}$ mice ('WT') and PBS- and DT-treated ApoE-RMB mice. ***, p < 0.001, Kruskal Wallis test followed by Dunn's multiple comparison. (D) Macroscopic images of the hearts and aortas of DT- and PBS-treated mice displaying aneurysms (indicated by arrows). (E) Consecutive cross-sections stained with Masson Trichrome (top, scale bar: 500 μm) or with Sirius red (bottom; polarised light). Aorta outer diameter (F) and collagen density of the aortic wall (G) for mice presenting aneurysms (PBS: n = 7; DT; n = 6) were calculated by computer-assisted morphometry on cross-sections stained with Sirius red. Data from sections taken at different levels (every 300 μm) from each aneurysm were used for calculations (S8A Fig), and aligned to the layer with the largest haematoma. Mean +/- standard error; p-values for temperature effect in mixed-model (REML) analysis.

peritoneal cavity, whereas the basophil population had fully replenished the blood of ApoE-RMB mice (S7 Fig). As an attempt to assess of MC activity in the plasma could be used as a marker of AAA progression, we measured plasma levels of tryptase at the end of the procedure (Day 28 of AngII infusion); however, tryptase was undetectable in any mouse. Presence of aneurysms or treatment did not induce an increase in plasmatic levels of inflammatory cytokines (S7 Fig).

DT injection reduced the expansion of the aneurysm (Fig 4D and 4F), comprising the size of the intramural haematoma and of the perivascular adventitial cuff (S8A–S8C Fig). Furthermore, on the cross-sections of the dissected aorta segments stained with picrosirius red we observed that the aortic wall of the aneurysm of DT-treated mice presented an increased density of collagen compared to control mice (Fig 4E and 4G), especially in the adventitial fibrous cuff (S8D Fig). Our data therefore suggest that MC depletion improves the adventitia fibrotic remodelling which is critical for providing an adequate stiffness and strength subsequently constraining the expansion of the haematoma and the reduction of the aneurysmal progression [33].

## Discussion

This study reveals the existence of a potential amplification loop involving IgE[+] TLO B cells and MCs at site of tissue fissuring, which could constitute an inflection point in AAA evolution, precipitating aneurysmal progression (S1 Graphical abstract).

### Graphical abstract

Locally produced IgEs, complexed with lesion-specific antigens, activate MCs and trigger a self-sustained loop involving TLO B cells that may drive the progression of dissecting AAAs. MC-derived IL-4 promotes the synthesis of more IgEs, which exacerbates the pathology, potentially leading to AAA rupture. These findings highlight the importance of investigating the involvement of the IgE/MC/B cell axis in the development and progression of AAA, and may inform the development of novel therapeutic strategies targeting this pathway.

To our knowledge, this is the first time IgE[+] germinal centre B cells have been reported in TLOs. Indeed, while they appear early in primary immune responses, coinciding with the peak of IL-4 production, IgE[+] cells are unable to populate the long-lived B cell compartment, which explains why IgE[+] germinal centres are rarely seen in lymphoid organs [34]. Our data show that the observed IgE[+] TLOs develop close to aortic wall fissures. Strikingly, the proximity of MCs and IgE[+] TLOs at site of tissue injury in the wall of progressive aneurysms supports a possible role for these highly reactive inflammatory cells in the progression of the disease and in the maintenance of the pathologic loop, through their production of IL-4 in response to locally produced IgEs, which in turn sustains the IgE[+] germinal centre cells in local TLOs. The correlation observed in the adventitia between IgE and IL-4 concentrations, as well as between IL-4 and tryptase concentrations, provides compelling evidence for the interrelated activities of MCs and B cells. It is worth noting that other cell types besides MCs may also produce IL-4 and respond to IgEs [13, 35]. Interestingly, IL-4 may play opposing roles at different stages of the pathology. Experimental models suggest that IL-4 produced by eosinophils may protect against the development of aneurysms [35]. However, our data suggest that IL-4 may accelerate the rupture of established aneurysms.

Furthermore, it should be emphasized that the ROSA human MC line requires cross-linking of purified IgEs by an anti-IgE antibody for activation [24]. However, we did not induce such cross-linking with the conditioned medium. The activation of MCs under these experimental conditions suggests that the IgEs present in the conditioned medium were likely cross-

linked by their cognate antigens. In contrast, our data indicate that blood IgEs (not shown) failed to activate MCs, suggesting that they were not circulating as immune complexes. This strongly suggests that the antigen recognized by IgEs may be specifically present in the aneurysmal aorta. Additionally, locally produced IgEs may contribute to AAA progression by promoting the senescence of smooth muscle cells [36]. IgEs possess another potential function that could participate in aneurysm progression, which is to promote the survival of MCs. We did not explore the processes responsible for the elevated MC count in the diseased aorta. This phenomenon may primarily result from either recruitment, local proliferation or a better survival of MCs. As IgEs have been shown to prevent MC apoptosis [37, 38], they could be in part responsible for the accumulation of MCs in AAA aortas.

Whereas MC IL-4 production was dependent on IgEs present in the conditioned medium from aneurysmal aorta, degranulation was not. This indicates that besides IgEs, other molecules from the adventitia stimulate MCs. Indeed, other classes of immunoglobulins, which are also enriched in aneurysmal aortas [6], cytokines, DAMP-containing molecules and inflammatory mediators released by the injured vascular stroma could also trigger MC degranulation. Further studies focusing on the relationship between IgEs and MC activation could lead to novel therapeutic strategies to reduce chronic inflammatory disorders.

Our pre-clinical *in vivo* data directly support a role for MCs in the fate of aneurysmal development of arterial wall subjected to fissuring and formation of an intramural haematoma. Importantly, for reaching this conclusion, we used the ApoE-RMB mouse model in which the depletion of MCs, conditioned by the use of DT, started only after the initial trigger (aortic dissection). Instead, the rodent models previously used to address this question were constitutively defective in MCs [14, 16, 18], implying that the deficiency of MCs preceded, and possibly biased, the experimental intervention itself. Furthermore, constitutive MC-deficient models such as $Kit^{W/Wv}$ and $Kit^{W-sh/W-sh}$ mice display other hematopoietic abnormalities, notably neutrophilia, thrombocytosis, and macrophage defects [22, 39] which could be responsible for some of the effects attributed to MCs. In addition, as resident MCs are present in healthy aortas, constitutive MC depletion could impact aorta homeostasis as well as the formation/progression/rupture of AAAs in these models. By triggering MC depletion after the occurrence of dissections in our ApoE$^{-/-}$ RMB mouse model, we were able to specifically tackle the role played by MCs during the healing processes associated with the progression of AAAs in this model. We found that AAAs in MC-depleted animals displayed improved arterial remodelling with an increased collagen content of the aneurysmal wall, supporting an active role for MCs in collagen degradation and expansion of AAA [40, 41]. Mechanistically, MC chymase and tryptase could increase collagen degradation by converting collagenases from their inactive to their active form [16]. MCs could also inhibit the production of collagen by smooth muscle cells [42]. Hence, MCs could actively contribute to weaken the arterial wall at sites of micro-fissures and favour the iterative expansion of aneurysms, under the biomechanic stress of intraluminal thrombus [43, 44]. Furthermore, the unhealed micro-fissures could allow the accumulation of plasma proteins into the arterial wall, which is associated to AAA growth [45].

Importantly, the experimental setting we used here is similar to the clinical settings of a recently undertaken clinical trial investigating whether pemirolast, a histamine H1 antagonist acting as a MC stabilizer in allergic conditions, could retard the growth of medium-sized AAAs [21]. This trial in which the major end-point was the change in aortic diameter as assessed by ultra-sound imaging, failed to report a significant improvement in the patients receiving the MC inhibitor. Although this study was elegantly designed and performed, it did not however bring a definitive conclusion regarding MC involvement in AAA pathophysiology because i/ the pemirolast treatment failed to decrease the plasma tryptase levels, indicating

a critical dosing issue, ii/ the study interval might have been too short, and iii/ contrary to our study, the effect that MC inhibition might have exerted on tissue composition could not be studied in this clinical study. Hence, in light of our results, we believe it would be interesting to re-evaluate the effect of perimolast (and/or other MC-inhibiting drugs) on AAA growth and rupture rate by targeting patients presenting elevated levels of IgEs and focusing on adventitial remodelling. Inhibiting other key steps of the amplification loop, for instance by using anti-IgE antibodies, could be another promising therapeutic strategy for such patients. Finally, it would be interesting to evaluate if levels of circulating IgEs, tryptase, or IL-4 correlate with aneurysm growth in a longitudinal study, and therefore be used as markers for AAA progression.

The present study has several limitations that should be acknowledged. First, our conclusion on the pathological role of a MC and B cell amplification loop in AAA is based on observations of human samples and experimental models. Intervention studies are needed to formally demonstrate the involvement of such a loop in patients. Second, in our experimental mouse model, basophils are also depleted, even if transiently, which means that an involvement of basophils in the fibrotic remodelling in the adventitia and the progression of aneurysms cannot be ruled out. Finally, the aneurysm model of AngII infusion in ApoE-/- mice lacks TLO in the adventitia of the aorta, which limits its ability to test the impact of a local amplification loop involving TLO B cells, even though it may still be useful in investigating the involvement of MCs in preventing aneurysm worsening.

In spite of the aforementioned limitations, our findings strongly suggest that a local amplification loop involving IgE, MCs, and TLO B cells plays a critical role in the development and progression of AAA. This self-sustaining loop, driven by repetitive MC degranulation, could constitute a turning point in the course of the disease, leading to impaired arterial healing and ultimately AAA rupture. Our study provides a valuable framework for further investigations into the pathological mechanisms underlying AAA and may inform the development of novel therapeutic strategies targeting this debilitating condition (resumed in S1 Graphical abstract).

## Supporting information

**S1 Table. Patients'characteristics.**
(PDF)

**S2 Table. NAA healthy donors'characteristics.**
(PDF)

**S3 Table. ApoE-RMB survival and aneurysm occurrence.**
(PDF)

**S1 Fig. IgEs are elevated in the plasma and adventitia of AAA patients.**
(PDF)

**S2 Fig. GC B cells and plasma cells are elevated in AAA adventitia.**
(PDF)

**S3 Fig. AAAs with ancient or no intramural haematomas.**
(PDF)

**S4 Fig. Proximity of micro-fissures and MC degranulation in human AAAs.**
(PDF)

**S5 Fig. MCs accumulate in the adventitia of human AAAs.**
(PDF)

**S6 Fig. No correlation between tryptase, IgE and IL-4 in the plasma of AAA patients.**
(PDF)

**S7 Fig. Repopulation of basophils in the blood and MCs in the peritoneum after DT depletion in ApoE RMB mice.**
(PDF)

**S8 Fig. MCs promote aneurysm expansion after dissection in ApoE-RMB mice.**
(PDF)

**S1 File. Antibodies.**
(XLSX)

**S1 Graphical abstract.**
(TIF)

## Acknowledgments

The authors thank Melanie Gettings for her help with manuscript editing, and Emilie Monnet for her help with the animals.

## Author Contributions

**Conceptualization:** Alexia Loste, Jean-Baptiste Michel, Pierre Launay, Giuseppina Caligiuri, Antonino Nicoletti, Marie Le Borgne.

**Data curation:** Catherine Deschildre.

**Formal analysis:** Alexia Loste, Marie Le Borgne.

**Funding acquisition:** Grégory Gautier, Pierre Launay, Giuseppina Caligiuri, Antonino Nicoletti, Marie Le Borgne.

**Investigation:** Alexia Loste, Marc Clément, Sandrine Delbosc, Kevin Guedj, Jean Sénémaud, Anh-Thu Gaston, Marion Morvan, Guillaume Even, Emanuele Procopio, Lydia Deschamps, Marie Le Borgne.

**Methodology:** Alexia Loste, Grégory Gautier, Pierre Launay, Giuseppina Caligiuri, Antonino Nicoletti, Marie Le Borgne.

**Resources:** Grégory Gautier, Alexander Eggel, Michel Arock, Catherine Deschildre, Liliane Louedec, Jean-Baptiste Michel, Yves Castier, Raphaël Coscas, Jean-Marc Alsac, Pierre Launay.

**Supervision:** Giuseppina Caligiuri, Antonino Nicoletti, Marie Le Borgne.

**Visualization:** Alexia Loste, Jean Sénémaud, Marie Le Borgne.

**Writing – original draft:** Alexia Loste, Giuseppina Caligiuri, Antonino Nicoletti, Marie Le Borgne.

**Writing – review & editing:** Alexia Loste, Marc Clément, Sandrine Delbosc, Kevin Guedj, Jean Sénémaud, Grégory Gautier, Alexander Eggel, Michel Arock, Emanuele Procopio, Catherine Deschildre, Liliane Louedec, Jean-Baptiste Michel, Lydia Deschamps, Yves Castier, Raphaël Coscas, Jean-Marc Alsac, Pierre Launay, Giuseppina Caligiuri, Antonino Nicoletti, Marie Le Borgne.

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
