## [Decision Letter · Decision Letter 0]

1 Aug 2023

PONE-D-23-12902Involvement of an IgE/Mast cell/B cell amplification loop in abdominal aortic aneurysm progressionPLOS ONE

Dear Dr. Marie Le Borgne,

Thank you for submitting your manuscript to PLOS ONE. After careful consideration, we feel that it has merit but does not fully meet PLOS ONE’s publication criteria as it currently stands. Therefore, we invite you to submit a revised version of the manuscript that addresses the points raised during the review process.

We look forward to receiving your revised manuscript.

Kind regards,

Venkateswaran Subramanian, Ph.D

Academic Editor

PLOS ONE

Journal Requirements:

"I have read the journal's policy and the authors of this manuscript have the following

competing interests: A Eggel is a cofounder and scientific advisor of Excellergy, INC.

and ATANIS Biotech AG. M. Arock is on DSMB for AB Science and advisory board for

Blueprint Medicines; receives consulting fees and/or honoraries from AB Science,

Blueprint Nedicines and Novartis; and declares patent #WO2013064639A1 ‘Human

mastocyte lines, preparation and uses. P Launay is the CEO of Inatherys."

3. Thank you for stating the following in the Competing Interests/Financial Disclosure * (delete as necessary) section:

"I have read the journal's policy and the authors of this manuscript have the following

competing interests: A Eggel is a cofounder and scientific advisor of Excellergy, INC.

and ATANIS Biotech AG. M. Arock is on DSMB for AB Science and advisory board for

Blueprint Medicines; receives consulting fees and/or honoraries from AB Science,

Blueprint Nedicines and Novartis; and declares patent #WO2013064639A1 ‘Human

mastocyte lines, preparation and uses. P Launay is the CEO of Inatherys."

We note that you received funding from a commercial source: " Excellergy, INC."

Reviewers' comments:

Reviewer's Responses to Questions

**Comments to the Author**

1. Is the manuscript technically sound, and do the data support the conclusions?

Reviewer #1: Yes

Reviewer #2: Partly

2. Has the statistical analysis been performed appropriately and rigorously? 

Reviewer #1: Yes

Reviewer #2: Yes

3. Have the authors made all data underlying the findings in their manuscript fully available?

Reviewer #1: Yes

Reviewer #2: Yes

4. Is the manuscript presented in an intelligible fashion and written in standard English?

Reviewer #1: Yes

Reviewer #2: No

5. Review Comments to the Author

Reviewer #1: The authors aimed to provide novel mechanisms insight the AAA progression by studying the role of MCs activation by IgEs released by TLOs located in the adventitia of human AAA. To achieve this objective they have exhaustively investigated the location of MCs and IgE producing cells in the AAA wall and their colocalization with the intraluminal haematomas. However, it is very difficult to demonstrate the cause (IgE secretion by lymphocytes B in TLOs) and effect (activation of MC) in human samples, if not impossible. I am aware of the difficulties of working with fresh human tissue to run assays which involve flow cytometry with the digested adventitia layer and the conditioned media of isolated adventitia. They additionally conducted an elegant in vitro study with ROSA human mast cells line to demonstrate the influence of conditioned medium from AAA and NAA adventitias on the IL4 production and degranulation process of MC including the use of anti-IgE protein to reverse the effect. Finally, they depleted MCs in an AAA murine model very well known to check if the intramural thrombus and the AAA incidence were affected.

This is a very well conducted work and a very clear and straight forward study, however, I found few questions that need to be addressed. My concerns are outlined below:

Comments:

1) I do not agree with the pseudoaneurysm terminology. Perhaps, you can talk of pseudoaneurysm where the intramural thrombus is formed in the abdominal aorta but the aneurysm formation can be observed in the flanking regions of the suprarenal abdominal aortic aneurysm in this murine model.

2) Why did the authors analysed IL-4 and Tryptase on a different set of patients?.

3) Did the authors perform the quantification of plasma IgE levels and IL4 in their AAA patients cohort to analyze if there were changes before surgery and post-surgery during the patients’ follow up?.

4) As the authors mentioned in discussion section, the AngII infusion in ApoE-/- mice lacks TLO in the adventitia of the aorta and for this reason they are not considered a good model to test the impact of a local amplification loop involving TLO+B cells. My question is, why the authors did not use another AAA murine model?.

Reviewer #2: This study by Loste et al investigates the involvement of an IgE/Mast cell/B cell amplification loop in the progression of abdominal aortic aneurysm (AAA). The authors attempt to identify the potential role and mechanism linking local mast cells (MCs), tertiary lymphoid organs (TLO) B cells and IgE production in AAA progression by examining human AAA tissues and angiotensin II (Ang II)-induced AAA mouse models. The authors conclude that interaction between MCs located close to aortic wall fissures and IgE+ B cells located in the adventitial TLO amplify AAA progression and rupture by activating MC and driving IL-4 production.

The role of MCs in AAA has been well studied in the past and there is compelling evidence suggesting that MCs play a role of the development of experimentally induced AAA. Since there are no medical drugs to treat AAA, MC inhibitors could be of value in treating AAA patients. The current manuscript to identify the mechanism of mast cell mediated AAA progression involving MC/B cell/IgG loop represent only a marginal increment to the current knowledge on the role of mast cells in AAA progression and does not provide sufficient mechanism to dissect these pathways. Although the study is of importance, manuscript in its current forms has the following concerns.

1. The important question is what causes the mast cells to migrate to the AAA lesions? This needs to be discussed.

2. How does the authors characterize the presence of TLOs? Is the composition of cells in TLOs differ in human patients versus mouse models? Does it contain other immune cells besides B cells?

3. Mast cells largely participates in the inflammation. Have the authors looked into the correlation of mast cells with inflammatory cytokines in the plasma of AAA patients? This is also important in the mouse model.

4. The presence of mast cells in the adventitia of AAA patients contradicts with the previous report where MCs were present in the medial layer.

5. The conditioned medium collected from adventitia of AAA tissues does not necessarily answer that activated MCs are the sole source of CD63 and IL4. The adventitia of AAA tissue is composed of many innate and adaptive immune cells in addition to vascular cells that contributes to activation of mast cells (Figure 3). Many other cells mainly activated T cells produce CD63 and IL4. And IL-4 is mostly anti-inflammatory and protective. The author needs to examine other cytokines and soluble factors that may be produced by mast cells.

6. Most of the findings are concluded from imaging analysis. It would be important to confirm such interaction between mast cells and B cells in in vitro cell culture model.

7. The use of mouse model conditionally lacking mast cells has added impact to the manuscript although previous studies have already identified the direct role of mast cells in AAA. No clear mice number has been mentioned. Is there a reason to use 28-week-old mice? Most literatures in the field have used 8-12 weeks old mice.

8. Also, it would be interesting if the authors could examine MC activity by measuring plasma level of circulating tryptase, chymase or cathepsin G in the mouse model with mast cell depletion. This will determine if such markers correlate with the progression or expansion rate of AAA and can be used as a biomarker to define progression.

9. There are minor errors in sentence structure and Figure citations that needs to be corrected.

6. PLOS authors have the option to publish the peer review history of their article (what does this mean?). If published, this will include your full peer review and any attached files.

Reviewer #1: No

Reviewer #2: No

---

## [Author Response · Author response to Decision Letter 0]

29 Sep 2023

We thank the reviewers for their careful examination of our manuscript and their helpful suggestions. Our answers to their comments are below.

Reviewer #1: 

The authors aimed to provide novel mechanisms insight the AAA progression by studying the role of MCs activation by IgEs released by TLOs located in the adventitia of human AAA. To achieve this objective, they have exhaustively investigated the location of MCs and IgE producing cells in the AAA wall and their colocalization with the intraluminal haematomas. However, it is very difficult to demonstrate the cause (IgE secretion by lymphocytes B in TLOs) and effect (activation of MC) in human samples, if not impossible. I am aware of the difficulties of working with fresh human tissue to run assays which involve flow cytometry with the digested adventitia layer and the conditioned media of isolated adventitia. They additionally conducted an elegant in vitro study with ROSA human mast cells line to demonstrate the influence of conditioned medium from AAA and NAA adventitias on the IL4 production and degranulation process of MC including the use of anti-IgE protein to reverse the effect. Finally, they depleted MCs in an AAA murine model very well known to check if the intramural thrombus and the AAA incidence were affected.

This is a very well conducted work and a very clear and straight forward study, however, I found few questions that need to be addressed. My concerns are outlined below:

Comments:

1) I do not agree with the pseudoaneurysm terminology. Perhaps, you can talk of pseudoaneurysm where the intramural thrombus is formed in the abdominal aorta but the aneurysm formation can be observed in the flanking regions of the suprarenal abdominal aortic aneurysm in this murine model.

Response: We were reluctant to call the pathology in Ang-II model as ‘aneurysm’, as there is no luminal enlargement, as is usually the case for actual aneurysms. Some authors had described the pathology as pseudoaneurysm, which is why we used it (doi: 10.1172/jci.insight.155815). However, we realize that this is controversial and therefore have replaced the term pseudoaneurysm by aneurysm in the text.

2) Why did the authors analysed IL-4 and Tryptase on a different set of patients?

Response: Conditioned medium from a set of patients was used for dosage of IgE and tryptase, as well as for experiments looking at the effect of conditioned medium on mast cells in vitro (Figure 3). These experiments consumed the condition medium for several patients, and the leftovers from the other samples had been thawed and frozen repetitively, and were therefore unsuitable to conduct cytokine dosage. This is why the analysis of IL-4 and tryptase was conducted on a different set of patients. This is now explained in the Materials and Methods section of the revised manuscript.

3) Did the authors perform the quantification of plasma IgE levels and IL4 in their AAA patients cohort to analyze if there were changes before surgery and post-surgery during the patients’ follow up?.

Response: The question raised is indeed pertinent; the measurement of plasma IgE and IL-4 levels before and after surgery would be highly insightful. Regrettably, our authorization from the ANSM and CPP did not grant us access to the patients after the surgery, as our work was exploratory. In the future, this work could be used as the justification to conduct such a medical study and get approval from the relevant regulatory authorities. This point is mentioned in the discussion of the revised manuscript : “Finally, it would be interesting to evaluate if levels of circulating IgEs, tryptase, or IL-4 correlate with aneurysm growth in a longitudinal study, and therefore be used as markers for AAA progression.”

4) As the authors mentioned in discussion section, the AngII infusion in ApoE-/- mice lacks TLO in the adventitia of the aorta and for this reason they are not considered a good model to test the impact of a local amplification loop involving TLO+B cells. My question is, why the authors did not use another AAA murine model?

Response: The Angiotensin II infusion model indeed induces aneurysms but does not lead to the development of TLOs. To the best of our knowledge, there is no existing mouse model for aneurysms that also includes TLO development. Both in humans and mouse models, TLOs typically develop in response to sustained inflammatory conditions. While Angiotensin II has the potential to increase blood pressure, a key factor in aneurysm development, the model's duration is insufficient to establish the chronic inflammation necessary for TLOs to appear. Unfortunately, although our aim was to develop a model to investigate the interaction between mast cells and TLOs B cells in aneurysm development, it appears that the mouse model may not be suitable for addressing this question. However, even though the model may not be ideal for elucidating how the dialogue between mast cells and IgEs producing TLO B cells exacerbates aneurysms by potentially causing new microtears in the aortic wall, it did allow us to examine the role of mast cells in the healing process of these microtears. 

Reviewer #2:

This study by Loste et al investigates the involvement of an IgE/Mast cell/B cell amplification loop in the progression of abdominal aortic aneurysm (AAA). The authors attempt to identify the potential role and mechanism linking local mast cells (MCs), tertiary lymphoid organs (TLO) B cells and IgE production in AAA progression by examining human AAA tissues and angiotensin II (Ang II)-induced AAA mouse models. The authors conclude that interaction between MCs located close to aortic wall fissures and IgE+ B cells located in the adventitial TLO amplify AAA progression and rupture by activating MC and driving IL-4 production.

The role of MCs in AAA has been well studied in the past and there is compelling evidence suggesting that MCs play a role of the development of experimentally induced AAA. Since there are no medical drugs to treat AAA, MC inhibitors could be of value in treating AAA patients. The current manuscript to identify the mechanism of mast cell mediated AAA progression involving MC/B cell/IgG loop represents only a marginal increment to the current knowledge on the role of mast cells in AAA progression and does not provide sufficient mechanism to dissect these pathways. Although the study is of importance, manuscript in its current forms has the following concerns.

Response: While many studies have been performed to understand the involvement of MC in AAA formation, their role in the progression of the disease was not addressed as far as we know. Furthermore, we believe that the mechanism that we unraveled where B cells that mature in the adventitial TLO mutually cooperate with MCs through their IgE production and IL-4, respectively, is new. This incremental knowledge cannot be considered as marginal since it provides a rational to envision a therapeutic strategy to block MC activation in patients specifically displaying heightened IgE responses.

1. The important question is what causes the mast cells to migrate to the AAA lesions? This needs to be discussed.

Indeed, the initial premise of our manuscript hinges on the observation that mast cell numbers increase in the diseased aorta, a phenomenon previously documented by other research groups (17, 18). Mast cells, being sentinel immune cells, naturally reside in various tissues, including the aortic wall, under steady-state conditions. Our manuscript, however, does not delve into the specific question of whether their increase is due to local proliferation or recruitment. To our knowledge, the precise mechanism underlying their expansion in AAAs remains unestablished and constitutes a crucial area for further investigation.

Response: To raise this question in our manuscript, we added the following sentence in the discussion “IgEs possess another function that could participate in aneurysm progression, which is to promote the survival of MCs. We did not explore the processes responsible for the elevated MC count in the diseased aorta. This phenomenon may primarily result from either recruitment, local proliferation or a better survival of MCs. As IgEs have been shown to prevent MC apoptosis (https://doi.org/10.1016/S1074-7613(01)00159-5, DOI: 10.1038/nri914), they could be in part responsible for the accumulation of MCs in AAA aortas.”

2. How do the authors characterize the presence of TLOs? Is the composition of cells in TLOs differ in human patients versus mouse models? Does it contain other immune cells besides B cells?

Response: We have detected the presence of tertiary lymphoid organs (TLOs) through flow cytometry and/or immunofluorescent staining, when germinal center B cells (GCBCs) are observed in non-lymphoid tissues. By flow cytometry, we identified GCBC as cells exhibiting the following markers: IgD- CD38med, HLA-DR+ (see fig. S2). In histological sections, TLOs were identified as DAPI+ CD19+ B cells organized into dark and light zones (see fig. 1 & 2). 

In previous studies, careful characterizations of these structures were conducted to confirm that they presented other characteristics of tertiary lymphoid organs. For instance, thorough characterization of TLOs in AAA adventitia were previously published by our laboratory (https://doi.org/10.1161/CIRCULATIONAHA.114.010988, https://doi.org/10.1371/journal.pone.0116295). Other immune cells are present besides B cells, such as CD4+ T cells (especially follicular helper T cells) and CD8+ T cells. Other structures/cells associated with germinal centers are also observed, such as high endothelial venules and follicular dendritic cells. Finally, other immune cells can be present in TLOs in different diseases, such as dendritic cells, macrophages, and innate lymphoid cells, and could be present in adventitial TLO also (https://doi.org/10.3389/fimmu.2021.674565).

Regarding the cellular composition of TLOs in humans versus mice, GC B cells, follicular DCs, HEVs and follicular helper CD4+ T cells can be observed also in mouse models (https://www.ahajournals.org/doi/10.1161/CIRCULATIONAHA.114.010988, https://rupress.org/jem/article/206/1/233/54206/Lymphotoxin-receptor-signaling-promotes-tertiary ). However, we paid special attention to B cells, T cells, and stromal cells. Therefore, discrepancies between the two organisms cannot be ruled out. For instance, a study was conducted to compare pancreatic TLOs from a pancreatic type 1 diabetes model and from human diabetic patients (https://doi.org/10.1007/s00125-021-05453-z); although largely similar, the authors could detect some differences, such as the presence of regulatory T cells in mouse but not human samples.

For a better understanding of the cellular composition and structures characterizing TLOs, we have amended the introduction with the following sentence: “These structures resemble germinal centres from secondary lymphoid organs, as they exhibit a highly organized structure with T- and B-cell areas, lymphatic vessels, high endothelial venules, follicular dendritic cells, and fibroblastic reticular-like cells, both in human samples and mouse model (DOI: 10.2119/molmed.2015.00027).”

3. Mast cells largely participate in the inflammation. Have the authors looked into the correlation of mast cells with inflammatory cytokines in the plasma of AAA patients? This is also important in the mouse model.

Response: Patients with AAA have chronic inflammation, and mast cells participate in this inflammation. We have indeed looked for other cytokines in the plasma of patients of AAA when we looked for IL-4. Levels of cytokines were quite low (just above the lower detection limit of the kit for IL-6; no detectable amount of TNF-alpha or IL-10). There were correlations between the concentration of the different cytokines in the plasma (suggesting that the levels that we measured, even if low, were true), but there was no correlation between any cytokine and tryptase. These data have been added to Figure S6. 

It should be noted that in our patients, the level of cytokines produced in the adventitia did not correlate with the level of cytokines present in the serum. This was also true for tryptase. The level of systemic inflammation therefore does not correlate with the level of local inflammation. We have added the graphs illustrating the absence of correlation for IL-4 and tryptase to Figure S6, and these data have been referred to in the revised manuscript.

In the mouse model, circulating tryptase (mcpt6) levels in the plasma were below the detection limit of the ELISA kit. We also measured cytokine levels by luminex, but the only one above background was IL-4 (no IL-6, IL-1beta, TNF-alpha detectable). IL-4 levels did not correlate with the presence or absence of aneurysm in mice (showed below), or with treatment. Therefore, as in Humans, the local inflammation in the aorta does not seem to be reflected in the circulation. Data for IL-4 has been added to Figure S7, and included in the result section.

4. The presence of mast cells in the adventitia of AAA patients contradicts with the previous report where MCs were present in the medial layer.

Response: Indeed, as referenced in the manuscript, prior research has demonstrated the increase in MCs number within the medial and adventitial layer of the aorta (https://www.ahajournals.org/doi/full/10.1161/CIRCRESAHA.108.173682). Our findings align with this observation, and we have illustrated some MCs within the medial layer of the aorta in Figures S4 and S5 of our manuscript. However, in the samples that we have examined, MCs are primarily situated within the adventitia.

5. The conditioned medium collected from adventitia of AAA tissues does not necessarily answer that activated MCs are the sole source of CD63 and IL4. The adventitia of AAA tissue is composed of many innate and adaptive immune cells in addition to vascular cells that contributes to activation of mast cells (Figure 3). Many other cells mainly activated T cells produce CD63 and IL4. And IL-4 is mostly anti-inflammatory and protective. The author needs to examine other cytokines and soluble factors that may be produced by mast cells.

Response: We agree with the reviewer that the source of IL-4 measured in the condition media from adventitias might be produced by other cells. This is indicated in the discussion: “It is worth noting that other cell types besides MCs may also produce IL-4 and respond to IgEs (13, 35). Interestingly, IL-4 may play opposing roles at different stages of the pathology. Experimental models suggest that IL-4 produced by eosinophils may protect against the development of aneurysms (35). However, our data suggest that IL-4 may accelerate the rupture of established aneurysms.” 

As pointed by the reviewer and stated in the introduction (“activated mast cells (MCs) release potent proteases, cytokines and vasoactive molecules, such as leukotriene and histamine”), mast cells are known to produce other cytokines, such as TNF-alpha, IL-1, IL-6 (https://doi.org/10.1111/imr.12634 ). In the conditioned media from adventitial aortas, we could detect the presence of IL-6, IFN-gamma and IL-17. Besides IL-4, tryptase levels correlated with levels of IFN-gamma, but not with IL-6 or IL-17. Of note, there was also a strong correlation between IL-4 and IFN-gamma, which was quite puzzling considering their known antagonist effects. This data was added to Figure S6, and stated in the result section. As IFN-gamma is not a major cytokine for B cell proliferation or differentiation toward IgE-producing B cells, we did not look if conditioned media from AAA adventitias induced IFN-gamma transcription in mast cells. 

6. Most of the findings are concluded from imaging analysis. It would be important to confirm such interaction between mast cells and B cells in in vitro cell culture model.

Response: A significant portion of the manuscript relies on descriptive histological observations. Nevertheless, we also conducted in vitro experiments to investigate the impact of IgE-antigen (Ag) complexes derived from diseased aorta on MC degranulation and IL-4 production. We were also eager to perform co-culture experiments with B cells and MCs to assess the MCs' ability to induce IgE production in B cells and the capacity of IgE-antigen complexes to stimulate MC degranulation and IL-4 production. However, executing such an experiment presented several challenges. First, a consequent amount of biologically active adventitial B cells had to be isolated from aneurysmal tissues. To do so, we did attempt to sort B cells, but numbers of alive GC B cells were too low to proceed with coculture experiments. Second, even if we could have obtained enough AAA B cells to co-culture them with mast cells, we would have needeed to add the correct antigen(s) against which IgEs are directed. These antigens are yet unknown and we were unsuccessful in isolating Ags from their complexes with immunoglobulins in the conditioned medium to identify or use them. Another possibility would be to add the conditioned medium, but its content in cytokines, chemokines, and DAMPs produced by many different cell types would have biased the results. Given these methodological hurdles, we could not run such experiments. 

7. The use of mouse model conditionally lacking mast cells has added impact to the manuscript although previous studies have already identified the direct role of mast cells in AAA. No clear mice number has been mentioned. Is there a reason to use 28-week-old mice? Most literatures in the field have used 8-12 weeks old mice.

Response: numbers are indicated in Table S3.

We chose the angiotensin II infusion model for a duration of 4 weeks into 28 weeks old ApoE-/- mice to conduct our project. 28-week old animals is the usual age of the mice in this model, except if the animals are put on a high-fat diet. This model best recapitulates the pathology we observed in humans: it develops against a backdrop of atherosclerosis, can lead to aortic rupture or dissections, exhibits an intraluminal thrombus, medial degeneration, and leukocyte infiltration (DOI: 10.1161/ATVBAHA.116.308534 ). 

8. Also, it would be interesting if the authors could examine MC activity by measuring plasma level of circulating tryptase, chymase or cathepsin G in the mouse model with mast cell depletion. This will determine if such markers correlate with the progression or expansion rate of AAA and can be used as a biomarker to define progression.

Response: Tryptase could not be detected in the plasma of the mice at the time of euthanasia, regardless of the presence of aneurysm and treatment (DT or PBS). This information was added in the result section. “As an attempt to assess of MC activity in the plasma could be used as a marker of AAA progression, we measured plasma levels of tryptase at the end of the procedure (Day 28 of AngII infusion); however, tryptase was undetectable in any mouse.”

It would have been interesting to check tryptase levels at earlier time points, but we did not collect plasma samples before sacrifice, so we cannot check if activation of mast cells in the aorta during the development of aneurysms can be detected in the plasma. 

9. There are minor errors in sentence structure and Figure citations that needs to be corrected.

Response: We corrected these errors as best as we could. The manuscript was corrected by English native speakers to avoid errors in sentence structure.

Journal requirements

Response: We edited the revised manuscript to meet PLOS ONE’s style requirements. 

Response: "I have read the journal's policy and the authors of this manuscript have the following competing interests: A Eggel is a cofounder and scientific advisor of Excellergy, INC. and ATANIS Biotech AG. M. Arock is on DSMB for AB Science and advisory board for Blueprint Medicines; receives consulting fees and/or honoraries from AB Science, Blueprint Nedicines and Novartis; and declares patent #WO2013064639A1 ‘Human mastocyte lines, preparation and uses. P Launay is the CEO of Inatherys."

Response: We included our Competing Interests statement in our cover letter.

3. Thank you for stating the following in the Competing Interests/Financial Disclosure * (delete as necessary) section:

"I have read the journal's policy and the authors of this manuscript have the following competing interests: A Eggel is a cofounder and scientific advisor of Excellergy, INC. and ATANIS Biotech AG. M. Arock is on DSMB for AB Science and advisory board for Blueprint Medicines; receives consulting fees and/or honoraries from AB Science, Blueprint Nedicines and Novartis; and declares patent #WO2013064639A1 ‘Human mastocyte lines, preparation and uses. P Launay is the CEO of Inatherys."

We note that you received funding from a commercial source: " Excellergy, INC."

Within this Competing Interests Statement, please confirm that this does not alter your adherence to all PLOS ONE policies on sharing data and materials by including the following statement: "This does not alter our adherence to PLOS ONE policies on sharing data and materials.” (as detailed online in our guide for authors http://journals.plos.org/plosone/s/competing-interests). If there are restrictions on sharing of data and/or materials, please state these. Please note that we cannot proceed with consideration of your article until this information has been declared. Please include your amended Competing Interests Statement within your cover letter. We will change the online submission form on your behalf.

Response: We did not receive funding from Excellergy, INC for the research presented in this manuscript. A Eggel is a cofounder and scientific advisor of Excellergy, INC, but this company did not provide funding for this study, or participate in any way in Study design, data collection and analysis, decision to publish or preparation of the manuscript. Our original financial disclosure stated : 

‘This work was supported by the Institut National de la Santé et de la Recherche Médicale (INSERM), the Université Paris Cité, and an Emergence grant from the Département Hospitalo-Universitaire ‘Fibrosis, Inflammation, REmodeling in cardiovascular, respiratory and renal diseases’ (DHU FIRE, Paris, France). AL was supported by the Domaine d’Intérêt Majeur ‘Maladies Cardiovasculaires, Obésité, Rein, Diabète’ (CORDDIM) from the Region Ile de France, and the Groupe de Réflexion sur la Recherche Cardiovasculaire (GRRC)/Fédération Française de Cardiologie (FFC, Paris, France). The funders had no role in study design, data collection and analysis, decision to publish, or preparation of the manuscript.

Response: We included the data that we referred to in Figures 4C and S7 of the revised manuscript.

---

## [Decision Letter · Decision Letter 1]

22 Nov 2023

Involvement of an IgE/Mast cell/B cell amplification loop in abdominal aortic aneurysm progression

PONE-D-23-12902R1

Dear Dr. BORGNE,

We’re pleased to inform you that your manuscript has been judged scientifically suitable for publication and will be formally accepted for publication once it meets all outstanding technical requirements.

Kind regards,

Venkateswaran Subramanian, Ph.D

Academic Editor

PLOS ONE

Reviewers' comments:

Reviewer's Responses to Questions

**Comments to the Author**

1. If the authors have adequately addressed your comments raised in a previous round of review and you feel that this manuscript is now acceptable for publication, you may indicate that here to bypass the “Comments to the Author” section, enter your conflict of interest statement in the “Confidential to Editor” section, and submit your "Accept" recommendation.

Reviewer #1: All comments have been addressed

Reviewer #2: All comments have been addressed

2. Is the manuscript technically sound, and do the data support the conclusions?

Reviewer #1: Partly

Reviewer #2: Yes

3. Has the statistical analysis been performed appropriately and rigorously? 

Reviewer #1: N/A

Reviewer #2: Yes

4. Have the authors made all data underlying the findings in their manuscript fully available?

Reviewer #1: Yes

Reviewer #2: Yes

5. Is the manuscript presented in an intelligible fashion and written in standard English?

Reviewer #1: Yes

Reviewer #2: Yes

6. Review Comments to the Author

Reviewer #1: The authors correctly answered my questions and fulfilled my requests in the reviewed version of the manuscript.

Reviewer #2: The authors have successfully addressed all the queries. There are no further comments on the manuscripts.

7. PLOS authors have the option to publish the peer review history of their article (what does this mean?). If published, this will include your full peer review and any attached files.

Reviewer #1: **Yes: **María Galán

Reviewer #2: No

---

## [Editor Report · Acceptance letter]

24 Nov 2023

PONE-D-23-12902R1 

Involvement of an IgE/Mast cell/B cell amplification loop in abdominal aortic aneurysm progression 

Dear Dr. Le Borgne:

I'm pleased to inform you that your manuscript has been deemed suitable for publication in PLOS ONE. Congratulations! Your manuscript is now with our production department. 

Kind regards, 

on behalf of

Dr. Venkateswaran Subramanian 

Academic Editor

PLOS ONE